# Identification of the Antidepressant Function of the Edible Mushroom *Pleurotus eryngii*

**DOI:** 10.3390/jof7030190

**Published:** 2021-03-08

**Authors:** Yong-Sung Park, Subin Jang, Hyunkoo Lee, Suzie Kang, Hyewon Seo, Seoyeong Yeon, Dongho Lee, Cheol-Won Yun

**Affiliations:** 1School of Life Sciences and Biotechnology, Korea University, Anam-dong, Sungbuk-gu, Seoul 02841, Korea; dcomtrue@korea.ac.kr (Y.-S.P.); scvxx@hanmail.net (S.J.); dlgusrn0524@naver.com (H.L.); sthe327@korea.ac.kr (S.K.); hyewon330@korea.ac.kr (H.S.); 2Department of Plant Biotechnology, College of Life Sciences and Biotechnology, Korea University, Seoul 02841, Korea; stopyear322@naver.com (S.Y.); dongholee@korea.ac.kr (D.L.); 3NeuroEsgel Co., Anam-dong, Sungbuk-gu, Seoul 02841, Korea

**Keywords:** antidepressants, *P. eryngii*, tryptamine, mushrooms, fungi

## Abstract

*Pleurotus eryngii* produces various functional molecules that mediate physiological functions in humans. Recently, we observed that *P. eryngii* produces molecules that have antidepressant functions. An ethanol extract of the fruiting body of *P. eryngii* was obtained, and the extract was purified by XAD-16 resin using an open column system. The ethanol eluate was separated by HPLC, and the fraction with an antidepressant function was identified. Using LC-MS, the molecular structure of the HPLC fraction with antidepressant function was identified as that of tryptamine, a functional molecule that is a tryptophan derivative. The antidepressant effect was identified from the ethanol extract, XAD-16 column eluate, and HPLC fraction by a serotonin receptor binding assay and a cell-based binding assay. Furthermore, a forced swimming test (FST) showed that the mice treated with purified fractions of *P. eryngii* exhibited decreased immobility time compared with nontreated mice. From these results, we suggest that the extract of *P. eryngii* has an antidepressant function and that it may be employed as an antidepressant health supplement.

## 1. Introduction

Edible mushrooms are a fungal species and source of various drugs and food supplements. To date, many molecules that have physiological functions have been identified from various mushrooms and are available commercially. In particular, the effects of immune activation [1,2], resistance to infection by microbial pathogens [3], and resistance to cancer [4] have been reported for mushrooms. Among the various functions of mushrooms, the Magic Mushroom (*Psilocybe semilanceata),* which is an illegal mushroom in the USA, has been reported to have an antidepressant function [5]. *P. semilanceata* produces “psylocybin”, which has hallucinogenic effects but is now undergoing clinical testing as a possible antidepressant drug [6,7]. Psylocybin shows high activity in patients with major depressive disorder and is more effective than the antidepressants currently used [8].

Antidepressant drugs are classified largely as tricyclic antidepressants [9], monoamine oxidase inhibitors (MAO inhibitors) [10], and selective serotonin reuptake inhibitors (SSRIs) [11]. SSRIs show relatively minor side effects and are associated with low drug resistance, and fluoxetine, which is known as Prozac, is a representative prescribed SSRI drug [12]. Generally, SSRIs block the reuptake of serotonin into neuronal cells, which makes it possible for sufficient serotonin to remain in the neurotransmission pathway [13,14]. However, numerous side effects of SSRIs have been reported, even though they are prescribed in many countries.

*Pleurotus eryngii* is an edible mushroom species in the *Pleurotus* genus and *Basidiomycota* phylum, and it has a unique structure that consists of thick and meaty stems with small caps [15]. *P. eryngii* is known as an edible and medicinal mushroom that has a variety of nutrient and bioactive molecules. In particular, *P. eryngii* has high amounts of dietary fibers and vitamins [16,17]. *P. eryngii* has a strong ability to uptake minerals for incorporation into organic compounds [18,19]. Furthermore, *P. eryngii* synthesizes diverse biomolecules that have pharmacological effects, such as lovastatin [11], pleureryn [20], and ribonuclease [21], which decrease cholesterol levels [22,23], exhibit antiviral effects [12,24], and exhibit immunomodulatory effects, respectively [25,26,27]. Recently, it was reported that eryngeolysin, which is a hemolysin produced by *P. eryngii*, showed cytotoxic effects on leukemia cells and exhibited antibacterial functions [28]. These reports indicate that *P. eryngii* may be a useful material to be developed as a functional food and medicine at a relatively low cost.

Recently, we identified an antidepressant function of *P. eryngii*, and this antidepressant function was investigated by receptor binding assays and animal experiments. In this study, we identify the functional molecules of *P. eryngii*, which may facilitate future research on antidepressants.

## 2. Materials and Methods

### 2.1. Ethanol Extraction of P. eryngii

First, 10 kg of *P. eryngii* was prepared and sliced into small pieces. Next, the sliced *P. eryngii* was placed in a dry oven at 60 °C for O/N. After the *P. eryngii* was dried completely, it was placed in a blender and ground sufficiently to make *P. eryngii* powder, which was placed in a plastic bottle with 5 volumes of 99.9% ethanol (Samchun Co., Seoul, Korea). After that step, the mixture was fixed to a rocker shaker and shaken for 24 h. The ethanol extract was centrifuged (13,000 rpm, 20 min), and only the supernatant was retained. The supernatant was placed in a dry oven at 60 °C to make an extract pellet. The pellet was resuspended in ultrapure purified water (Milli-Q, Co., Billerica, MA, USA) to make a liquid extract, and the liquid extract was filtered using 3M paper to obtain a highly transparent *P. eryngii* solution. The obtained *P. eryngii* solution was poured into a glass column containing XAD-16 (Sigma, Co., St. Louis, MO, USA) resin. The resin XAD-16 was activated by being incubated with 50 mM potassium phosphate buffer (pH 7.5) for more than 30 min. After the resin was washed, 99.9% ethanol was applied to the glass column, and dark red ethanol solution was purified. To remove the ethanol, a vacuum concentrator was used. The pellet was named the mixed sample, and the mixed sample was resuspended in ultrapure water (Milli-Q, Co., Billerica, MA, USA) and filtered using a Sep-Pak C18 filter (Waters, Co., Eugene, OR, USA) to remove fat-soluble substances and leave only water-soluble substances.

### 2.2. HPLC Analysis of the Mixed Sample and Purification of Molecules with SSRI Functions

The pellet was obtained through vacuum evaporation, resuspended in distilled water, and analyzed using RP-HPLC with a UV detector and a nonpolar C18 RP-HPLC column (Agilent Eclipse XDB-C18 (5 µm) 4.6 × 250 mm, Agilent Co., Santa Clara, CA, USA). The flow rate was 1 mL/min, and the mobile phase was methanol:water (50:50). Preparative HPLC was performed using a YMC C18 preparative column (10 × 250 mm) with the same gradient procedure as the analytical column.

### 2.3. FIA-MS Analysis

LC-MS analysis of the purified HPLC fraction was performed using liquid chromatography coupled with triple quadrupole mass spectrometry (TSQ Quantum ultra EMR, Thermo Fisher Scientific, San Jose, CA, USA) at the Korea Basic Science Institute (Seoul, Korea). The sample was analyzed with flow-injection analysis. A 5-µL sample was injected in a solvent (acetonitrile) stream into an ESI mass spectrometer. The MS spectrum was obtained in negative (positive) full scan mode (*m*/*z* 100–600). The mass parameters were as follows: spray voltage of 3 kV (4 kV), sheath gas pressure of 40 (arbitrary unit), aux gas pressure of 10 (arbitrary unit), and capillary temperature of 300 °C. For data acquisition and processing, Xcalibur v2.1.0 software (Thermo Fisher Scientific, San Jose, CA, USA) was utilized.

### 2.4. Plate Assay

For the *S. cerevisiae* growth assay, yeast strains with siderophore-specific deletion mutants were harvested from YPD (Yeast Peptone Dextrose) plates to distilled water for OD measurement. Next, 5 µL of yeast strains with OD550 values of 0.1, 0.01, and 0.001 was inoculated onto SD agar plates containing siderophore fractions or peaks with bathophenanthroline disulfonate (BPS). The plates were incubated at 30 °C for 1–3 days.

### 2.5. Yeast Strain, Media, and Growth Conditions

The yeast strains used in this study were wild-type YPH499 and Δfet3, Δfet3Δarn1, Δfet3Δarn2, Δfet3Δarn3, Δfet3Δarn1,2, Δfet3Δarn1,3, Δfet3Δarn1,4, Δfet3Δarn2,3, Δfet3Δarn3,4, Δfet3Δarn1,2,3, Δfet3Δarn1,2,4, Δfet3Δarn1,3,4, and Δfet3Δarn2,3,4 deletion mutant strains, which have been used in previous research [29]. Yeast strains were grown in YPD agar medium (1% (*w*/*v*) yeast extract, 2% (*w*/*v*) peptone, 2% glucose, and 2% agar powder). Iron-limited media were prepared with 100 µM bathophenanthroline disulfonate (BPS) and siderophores extracted from 20 µL P. eryngii or 20 µM ferrioxamine B (FOB). All cultures were incubated at 30 °C.

### 2.6. Cell Cultures

Two types of cell lines were employed in this study. The first cell line was AGS, derived from a human Caucasian gastric adenocarcinoma. The second cell line was HT29, derived from a human Caucasian colon adenocarcinoma. These cell lines were grown in RPMI 1640 medium. RPMI 1640 medium containing 10% FBS and 1% antibiotic-antimycotic is called RPMI 1640 [FBS+, A/A+]. RPMI 1640 that does not contain FBS and A/A is called RPMI 1640 [FBS−, A/A−]. DMEM that does not contain FBS and A/A is called DMEM [FBS−, A/A−]. All cell lines were cultured at 37 °C and 5% CO_2_.

### 2.7. MTT Assay

MTT assay was performed with the following procedure. First, 100 μL of cell solution was inoculated per well of a 96-well plate (AGS: 10^4^ cells/100 μL, HT29: 10^4^ cells/100 μL, HEK293: 3 × 10^4^ cells/100 μL). After inoculation, the plate was incubated for 24 h at 37 °C, the medium was subsequently removed from the well, and 100 μL of the samples with the medium were treated. The cells were incubated for 48 h, and the medium was subsequently removed. Next, 100 μL of 37 °C MTT medium was added to each well. After the treatment was completed, 100 μL of DMSO (Bio Basic, 67-68-5) was added to each well. Finally, a 96-well plate was placed in a plate reader and measured at a wavelength of 550 nm.

### 2.8. Receptor Binding Assay

Radiological binding assays were performed according to the protocols provided by the supplier of h5-HT1A human serotonin transporter proteins (PerkinElmer Inc., Boston, MA, USA). Briefly, ethanol extracts and HPLC fractions were diluted with an incubation buffer (50 mM Tris–HCl (pH 7.4), 5 mM KCl, and 120 mM NaCl), and the diluted samples were mixed with 14 µg of partially purified human serotonin transporter proteins and Paroxetin H^3^ (PerkinElmer Inc., Boston, MA, USA). Fluoxetine (10 mM) was used as a reference compound in this experiment. After incubation at 27 °C for 30 min, the reaction mixtures were filtered with a GFC filter unit followed by washing 5 times. The washed GFC filters were counted with a scintillation counter. These experiments were performed in duplicate.

### 2.9. Forced Swimming Test

The test was conducted as described earlier [30], with modifications. Individual mice were forced to swim for 6 min in plexiglass cylinders (30 cm in height with 20-cm internal diameters) filled with fresh water maintained at 25 ± 1 °C. Animals were assigned to different treatment groups (*n* = 8) after injection with either drug (treatment group) or vehicle (control group). Thirty minutes after i.p. administration of drugs at 20 mg/kg of mouse weight, individual mice were forced to swim for 6 min and the water was wiped off. Next, the mice were forced to swim again for 4 min in a test session, and immobility time was recorded. The immobility time of the treated mice was compared with that of the mice in the control group.

## 3. Results

### 3.1. Isolation of Siderophores from P. eryngii

First, we attempted to identify the siderophores produced by edible mushrooms and observed that *P. eryngii* produces siderophores. It has been reported that siderophores, which are iron chelators produced by most microorganisms, have anticancer effects, and we determined that siderophores are produced by the edible mushroom *P. eryngii*. To identify the siderophores produced by *P. eryngii*, we attempted to isolate them. The dried fruit body of *P. eryngii* was ground and extracted using ethanol, as shown in Figure 1, and the iron binding molecules were separated using an XAD-16 column (mixture). Next, the eluate from the XAD-16 column was separated by reverse-phase HPLC and, finally, the purified single HPLC fraction was analyzed using LC-MS.

The fruit body of *P. eryngii* was dried and extracted by ethanol for 24 h. In addition, it was then purified by XAD-16 column and HPLC. Finally, the well-isolated fraction of the HPLC chromatogram was subjected to structural analysis by LC-MS.

In each step of the purification procedure, the siderophores were analyzed qualitatively using yeast mutant collections that, as shown in Figure 2, showed growth defects, even though specific siderophores were supplied. We used *FET3* and *ARN* double-, triple-, and quadruple-deletion mutants, and each mutant showed growth defects, even when a specific siderophore was added to the media [29]. For example, *FET3*, *ARN1*, and *ARN3* triple-deletion mutants do not grow even when ferrichrome is added to the media because this mutant cannot use ferrichrome. The *FET3* and *ARN3* double-deletion mutants did not grow, even when ferrioxamine B was added to the media because this mutant could not use ferrioxamine B. Using these mutants, the HPLC fractions were tested to determine whether specific siderophores were included in each fraction. As shown in Figure 2A, fraction F2 exhibited growth defects in the only mutant that had the genotype of a *FET3*, *ARN1*, or *ARN3* triple-deletion mutant, but other mutants grew well. Therefore, we concluded that this fraction includes ferrichrome or ferrichrome derivatives. To purify the siderophores further, the F2 fraction was separated using HPLC, and we obtained 5 fractions, known as F2P1 through F2P5. The same plate assay with yeast deletion mutants was performed with those HPLC fractions, and F2P1 showed a growth pattern with F2 isolated previously. Furthermore, the F2P1 fraction was separated further, and we obtained three fractions, R1, R2, and R3. These fractions were subjected to yeast deletion mutant plate assays. As shown in Figure 2C, the F2P1R2 fraction (R2) showed the same phenotype as the F2P1 fraction, but the other two fractions showed different phenotypes than the F2P1 fraction. Finally, F2P1R2 was separated further, and we obtained 15 isolated peaks. Each well-isolated fraction was analyzed using the same yeast deletion mutants (Figure 2D). Fractions M1, M2, and M3 may have ferrichrome or ferrichrome derivatives, and we analyzed the structure of each fraction using LC-MS.

### 3.2. M4 Fraction of R2 Is Tryptamine

To analyze the structure of M1, M2, M3, and M4, we performed LC-MS analysis. As shown in Figure 2D, M4 occupied most of R2, and this finding led us to investigate M4 further. To identify the molecule of M4, we analyzed its exact structure. As shown in Appendix A, we analyzed the raw data of M4 using the MassBank site (www.massbank.jp, the Mass Spectrometry Society of Japan, Tokyo, Japan, 13 October 2020), and M4 was predicted to be tryptamine. Additionally, as shown in Figure 3A, the MS data showed a typical tryptamine structure, and these MS data were compared with commercial tryptamine using HPLC and MS. As shown in Figure 3B, the HPLC chromatogram showed that the M4 fraction and tryptamine exhibited the same retention time of 19.371 min, and the identity of the M4 fraction was further confirmed using MS analysis. As shown in Figure 3C, electrospray ionization (ESI)-mass spectrometry (ESI-MS) analysis indicated an *m*/*z* ratio of 157.59 for M4 and 157.41 for tryptamine. An ESI-MS analysis indicated that the entire spectrum of the *m*/*z* ratio of the M4 fraction corresponded to the spectrum of the *m*/*z* ratio of tryptamine. Tryptamine is a monoamine alkaloid and contains an indole ring structure with a molecular weight of 160.22 (Figure 3D). Tryptamine is synthesized from tryptophan and is observed from various plants and mushrooms, although its content varies depending on the organism. Tryptamine is expected to function as a neurotransmitter, and its derivatives are commercially available as antidepressant drugs. In this study, we attempted to purify the molecules or proteins that bind iron because we employed an XAD-16 column. However, tryptamine was isolated by the XAD-16 column purification step, and we investigated whether tryptamine binds to iron. Notably, we observed that tryptamine-HCl binds to iron, and the color was changed from colorless to brown by iron binding (data not shown). This finding explains why tryptamine was isolated from the XAD-16 column purification step. To determine the effects of tryptamine on depression, we investigated the function of the extract of *P. eryngii* as an antidepressant.

### 3.3. Extract of P. eryngii Has Antidepressant Activity

As described previously, the extract of *P. eryngii* contains tryptamine and, possibly, its derivatives, which implies that *P. eryngii* may be a candidate functional molecule for treating depression. To investigate the function of the *P. eryngii* extract in depression, we performed a receptor binding assay with the human serotonin receptor (HT1A, Perkin Elmer, Co, Downers Grove, IL, USA) to identify antidepressant activity using the Paroxetine-H^3^ (Perkin Elmer, Co., USA) radioisotope. As shown in Figure 4, fluoxetine (Prozac, Sigma–Aldrich, St. Louis, MO, USA) was employed as a positive control, and the competitive activity of the ethanol extract of *P. eryngii* was measured. The negative control was the reaction buffer itself, and the activity was calculated by comparison to the value of the reaction buffer. Fluoxetine (1 mM) inhibited 39% of the binding of paroxetine-H^3^ to the HT1A protein, and tryptamine (1 mM) inhibited 33% of the binding activity. Notably, 100 mg/mL ethanol extract of *P. eryngii* showed 68% inhibition of paroxetine-H^3^ binding, and these results indicate that *P. eryngii* has serotonin receptor binding activity. Next, we performed a receptor binding assay with the purified fraction of *P. eryngii*. XAD-16 column eluates of the *P. eryngii* extract (mixture) and R2 fraction of HPLC separation were employed in the receptor binding assay. The concentrations of the mixture and R2 were diluted from 0.01 mg/mL to 1 mg/mL, and the inhibition rate of receptor binding activity was measured. As shown in Figure 5A, the human serotonin receptor HT1A was used as a receptor protein, and fluoxetine was employed as a positive control. Fluoxetine (1 mM) inhibited 76% of receptor binding activity, and a mixture of 1 mg/mL fluoxetine inhibited 77% of receptor binding activity. In addition, 1 mg/mL R2 inhibited 91% of receptor binding activity, and the inhibition rate was concentration-dependent.

Furthermore, we tested whether the receptor-binding protein is upregulated in the cancer cell line and whether this effect is characteristic of AGS stomach cancer cells. We used the AGS cell line to perform a serotonin receptor binding assay, as shown in Figure 5B. The kinds of molecules and concentrations used in this binding assay were the same as in Figure 5A, and the inhibition rates of the mixture and R2 were similar to those in Figure 5A. Fluoxetine (1 mM) inhibited the receptor binding activity by 58%, and 1 mg/mL of the mixture and R2 inhibited the receptor binding activity by 37 and 71%, respectively. These results indicate that the mixture and R2 of *P. eryngii* have specific serotonin receptor binding activity.

### 3.4. Forced Swimming Test (FST) Showed the SSRI Activity of the P. eryngii Extract

To further characterize the antidepressant activity of *P. eryngii,* FST was performed. It is reported that the immobility time decreases in mice administered with SSRIs, and FST was performed to see the effects of these SSRIs. Therefore, it has been reported that the length of immobility time is related to the activity of SSRIs. The ethanol extracts of the *P. eryngii*, mixture, and R2 fractions were assayed by FST. In the FST, we performed two independent experiments with *n* = 4 in each group, as described in the Materials and Methods. As shown in Figure 6, the immobility time of the negative control injected with saline was 155 s with a 4-min swimming time. However, the immobility time was reduced to 118 s when fluoxetine was injected into mice at a dose of 20 mg/kg. In addition, 40-mg/kg doses of ethanol extract, mixture, and R2 fraction were injected, and the immobility time was measured. The mice treated with the ethanol extract exhibited an immobility time of 131 s, and the mixture and R2 showed immobility times of 124 and 107 s, respectively. Notably, the immobility time was decreased proportionally with the purification process of the *P. eryngii* extract. In other words, the antidepressant activity of the extract was increased proportionally with the purification process.

## 4. Discussion

Edible mushrooms are fungal species and have various nutrients that are beneficial to human health. These mushrooms also contain functional molecules for treating various diseases in humans, such as colon cancer, activating the human immune system, and serving as antibiotics against microbial pathogens [1,2,3,4]. Humans have been eating mushrooms for their health or as food for a long time without knowing the exact function of mushroom nutrients. Currently, researchers are focusing on identifying functional molecules from mushrooms as drug candidates, and many functional molecules have been developed as drugs. Recently, we also observed a novel function of the edible mushroom *P. eryngii*, and, in this report, we describe the antidepressant function of *P. eryngii.*

Depression is a mood disorder, and patients feel sadness, loss, and anger to an extent that interferes with normal human life. Depression can cause many problems in human activity at home and at work. Many factors affect depression, such as differences in neurotransmitters in the brain, genetic inheritance, personal differences, and diverse environmental factors [31,32,33]. Recently, many people have been diagnosed with depression, but adequate treatment is not always available, even though various drugs have been developed. Medicines for the treatment of depression is classified based on their modes of action, and these medicines primarily are selective serotonin reuptake inhibitors (SSRIs), serotonin and norepinephrine reuptake inhibitors (SNRIs), tricyclic antidepressants (TCAs), and monoamine oxidase inhibitors (MAOIs) [11]. Even though SSRIs have been the most commonly prescribed class of antidepressant medicine, some side effects for SSRIs have been reported, such as nausea, upset stomach, fatigue, headaches, dizziness, and insomnia [34,35,36]. Because of these side effects, new methods for the treatment of depression are in demand.

Natural compounds may represent another option for relieving depression, and it has been reported that various natural compounds from fish, plants, microorganisms, and mushrooms are effective against depression. For example, the omega-3 fats of fish may be involved in the functioning of serotonin in the brain. Additionally, nuts have high concentrations of omega-3 fats, and beans, seeds, vegetables, and probiotics are known to be helpful foods for depression [37,38,39]. However, the effect of whole foods on relieving depression is relatively low, and functional foods are needed to treat depression [40,41]. Recently, it has been reported that psilocybin, which is isolated from the Magic Mushroom *Psilocybe semilanceata,* shows psychedelic effects and has been proven to be an efficient therapy to treat major depressive disorder (MDD) in clinical tests [42]. Even though this mushroom is restricted from being sold as food in the United States, the active molecule psilocybin has been studied as a candidate drug to treat depression. Psilocybin was originally derived from tryptophan via tryptamine, dimethyltryptamine, and psilocin, and, finally, psilocybin was synthesized [43].

We also attempted to isolate functional molecules that affect depression. As described in the results section, the antidepressant molecule was identified by LC-MS analysis and was determined to be tryptamine and tryptamine derivatives. Tryptamine is a common functional group of biologically active compounds, such as neurotransmitters and psychedelic drugs, and is an intermediate of psilocybin biosynthesis [43]. Tryptamine is known to be an agonist of human trace amine-associated receptors (hTAARs) and acts as a nonselective serotonin receptor agonist [44,45]. Tryptamine also acts as a serotonin-norepinephrine-dopamine releasing agent (SNDRA) [46] and a noncompetitive inhibitor of serotonin N-acetyltransferase (SNAT) in mosquitoes [47]. However, tryptamine is rapidly metabolized by MAO-A and MAO-B; therefore, it has a very short half-life [48]. Tryptamine is synthesized from tryptophan by tryptophan decarboxylase [49]. Even though most plants and mushrooms synthesize tryptamine, its physiological function has not been thoroughly identified. In plants, tryptamine is a precursor for the synthesis of phytohormone indole-3-acetic acid [50,51] and monoterpene indole alkaloids, such as the anticancer agent vinblastine [52]. Therefore, the existence of tryptamine implies the existence of diverse tryptamine derivatives, such as neurotransmitters and other indole compounds. To date, tryptamine derivatives synthesized from *N*,*N*-dimethyltryptamine (DMT) and naturally occurring molecules have been reported, including 5-MeO-DMT (5-methoxy-*N*,*N*-dimethyltryptamine), 5-MeO-DPT (“Foxy-Methoxy,” 5-methoxy-*N*,*N*-dipropyltryptamine), AMT (2-(1H-indol-3-yl)-1-methyl-ethylamine), 4-AcO-DMT (4-acetoxy-*N*,*N*-dimethyltryptamine), and 4-AcODiPT (4-acetoxy-*N*,*N*-diisopropyltryptamine). Furthermore, it has been reported that gut microbiomes such as *Ruminococcus gnavus* and *Clostridium sporogenes* produce tryptamine in the mammalian gastrointestinal tract by decarboxylating dietary tryptophan [53] and help to regulate gastrointestinal electrolyte balance. Derivatives of tryptamine have been developed to treat migraine headaches [54].

In this study, we investigated functional molecules from *P. eryngii* that have antidepressant functions, and antidepressant activity was measured from mushroom extracts and purified fractions using HPLC analysis. The antidepressant activity was measured through in vitro and in vivo assays, and the activity was similar to that of commercially available fluoxetine, also known as Prozac. Notably, the crude extract of *P. eryngii* showed high antidepressant function in in vitro assays and animal tests. This result means that *P. eryngii* synthesizes tryptamine and its derivatives, and it is possible to use the extract itself to alleviate depression, thereby avoiding the toxicity of tryptamine encountered when using its purified form. It may be possible to co-administer antidepressant drugs to increase antidepressant drug activity.

## Figures and Tables

**Figure 1 jof-07-00190-f001:**
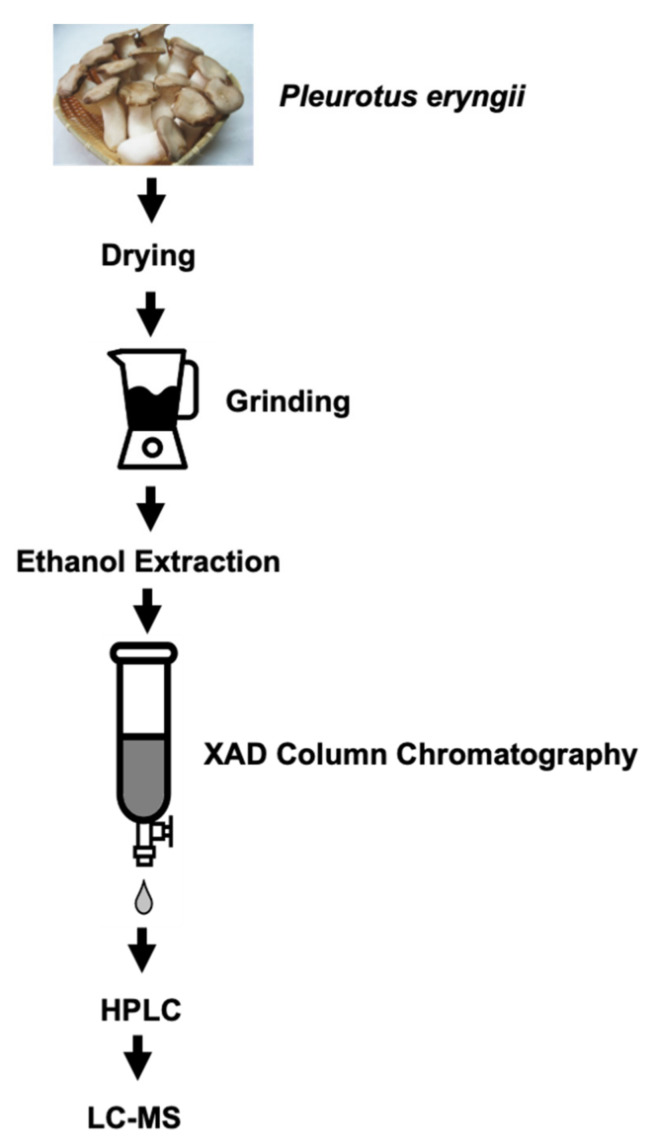
Procedure for isolating functional molecules from *P. eryngii*.

**Figure 2 jof-07-00190-f002:**
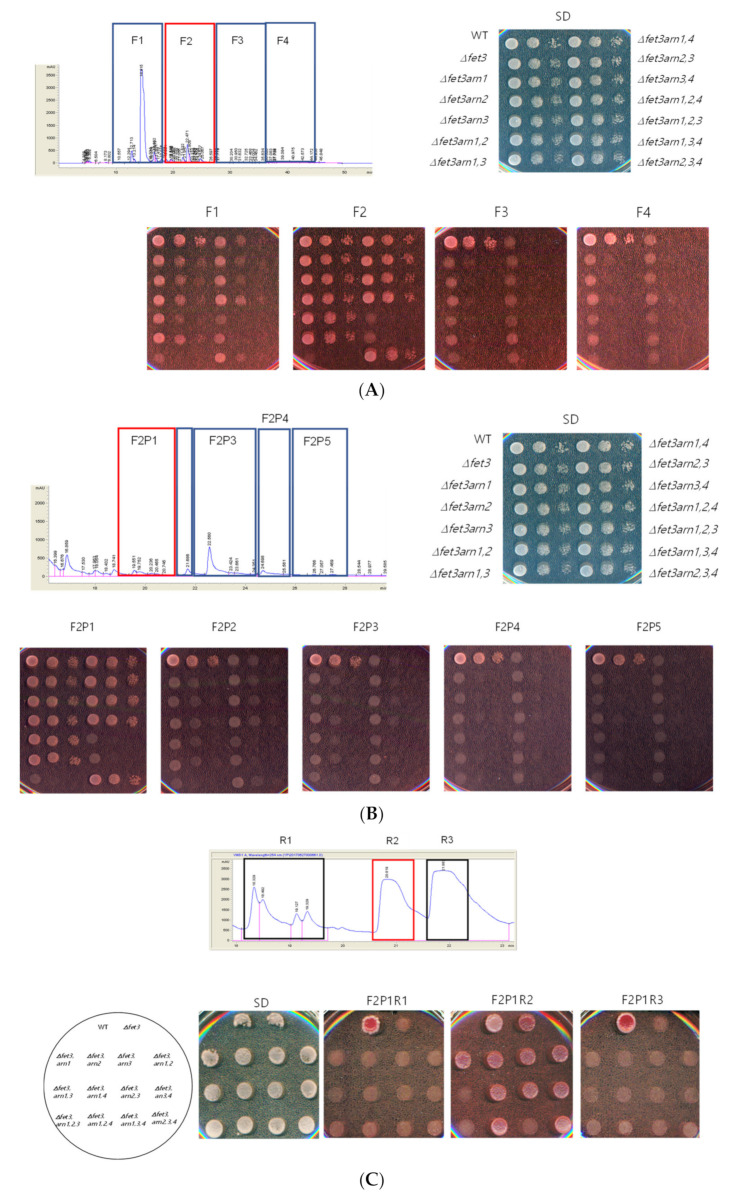
**HPLC analysis and purification of functional molecules from *P. eryngii.*** (**A**) The eluates of the XAD-16 column of the ethanol extract were separated by HPLC analysis into four fractions, F1, F2, F3, and F4. Then, each fraction was subjected to a plate assay to identify the fraction with ferrichrome. The genotype indicates the deletion mutant, as described in the Materials and Methods. Additionally, the plates used in this experiment are low iron media containing 10 mM iron and BPS. (**B**) The F2 fraction was separated further into five fractions: F2P1, F2P2, F2P3, F2P4, and F2P5. Each fraction was subjected to a plate assay to identify the fraction that had ferrichrome. (**C**) Additionally, the F2P1 fraction was separated into three fractions: F2P1R1, F2P1R2, and F2P1R3. Each fraction was subjected to a plate assay to identify the fraction that had ferrichrome. (**D**) Finally, the F2P1R2 fraction was separated into 15 fractions that showed well-isolated peaks, and each fraction was subjected to a plate assay to identify the fraction that had ferrichrome. The yeast genotypes inside the circle and square of C and D indicate the yeast strains used in this experiment.

**Figure 3 jof-07-00190-f003:**
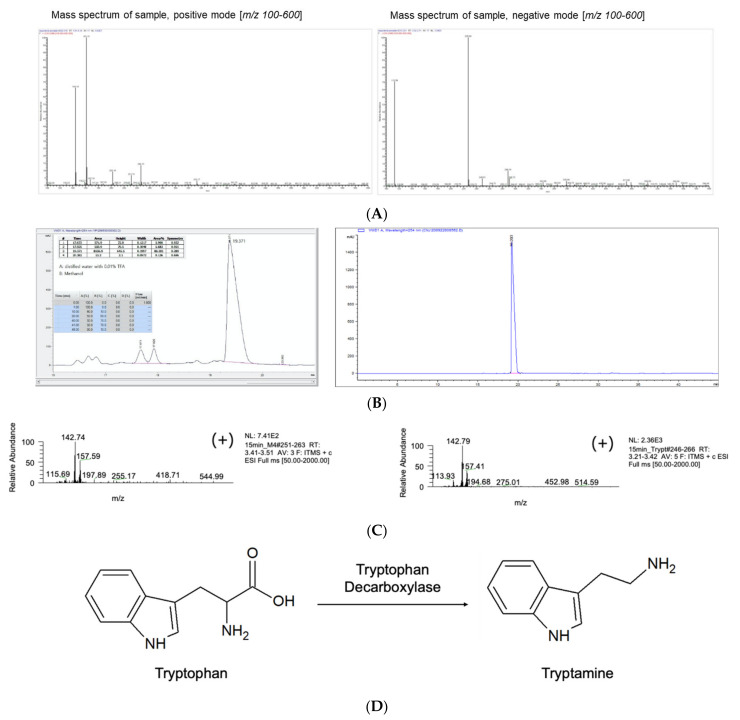
**M4 fraction is tryptamine**. (**A**) The fractions with ferrichromes and well-isolated peaks were analyzed by LC-MS to identify the molecular structure. In particular, the molecular structure of the M4 fraction was tryptamine. The mass spectrum showed typical tryptamine, and raw mass data showed that M4 was tryptamine. (**B**) To confirm the molecular structure of M4, commercial tryptamine (Sigma–Aldrich, St. Louis, MO, USA) was used as a standard molecule, and M4 and tryptamine were subjected to HPLC analysis. The left panel is M4 and the right panel is tryptamine. (**C**) Additionally, M4 and tryptamine were analyzed by LC-MS again for confirmation. The left panel is M4, the right panel is tryptamine, and the LC-MS chromatogram showed the same pattern. (**D**) The biosynthetic pathway to synthesize tryptamine from tryptophan.

**Figure 4 jof-07-00190-f004:**
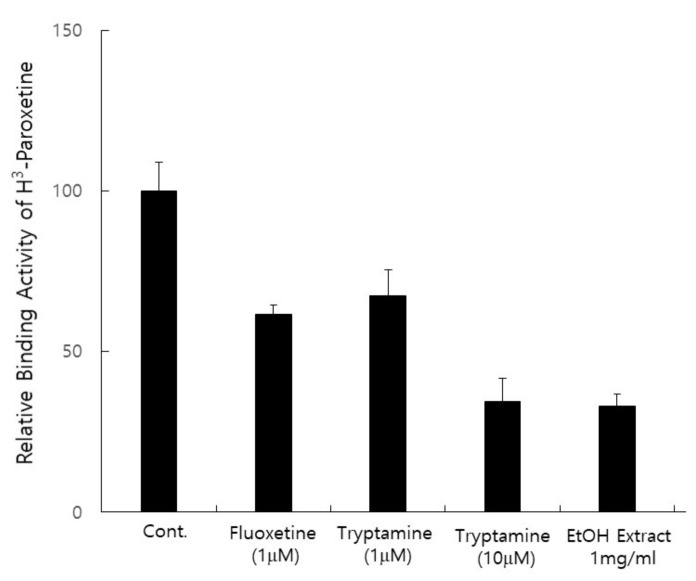
**Extract of *P. eryngii* showed inhibition activity of serotonin receptor binding.** The serotonin receptor binding assay was performed with an ethanol extract of *P. eryngii* to identify the neurotransmitter activity of the extract of *P. eryngii.* A 1 mg/mL concentration of *P. eryngii extract* was added to the reaction mixture of the serotonin receptor binding assay, and the inhibition rate of paroxetine-H^3^ binding to the human 5HT1-A protein was investigated. The positive control was 1 mM of fluoxetine (Prozac. Sigma–Aldrich, St. Louis, MO, USA), and tryptamine was also used to compare the activity with the extract of *P. eryngii*.

**Figure 5 jof-07-00190-f005:**
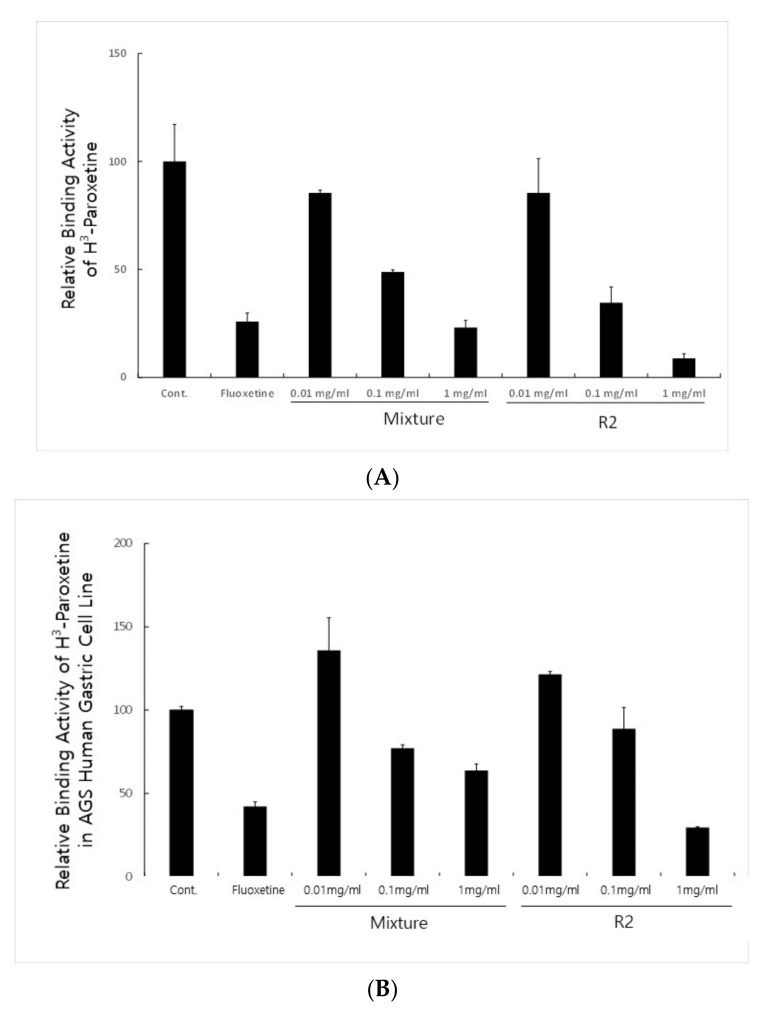
**Purified fractions of *P. eryngii* showed inhibition activity of serotonin receptor binding.** (**A**) The receptor binding assay was performed with the XAD-16 column eluates and R2 fraction of HPLC separation of *P. eryngii.* Fluoxetine was used as a positive control, and the negative control was buffer. The concentration of the XAD-16 column eluates and the R2 fraction was used from 0.01 mg/mL to 1 mg/mL. The experimental procedure and receptor protein used in this assay were the same as in Figure 4. The mixture indicates that the XAD-16 column eluates and R2 is F2P1R2. (**B**) Additionally, a receptor binding assay was performed with the AGS cell line, which is a gastric cancer cell line. The overall experimental procedure was the same as the receptor binding assay performed with the 5HT-1A protein in addition to using the cell instead of the protein.

**Figure 6 jof-07-00190-f006:**
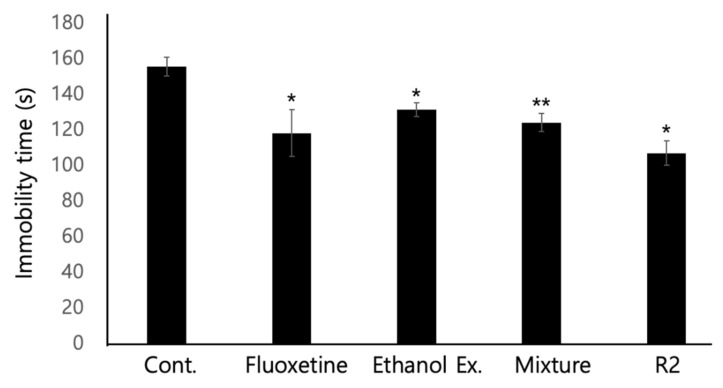
**Purified fractions of *P. eryngii* increased immobility time in****the FST experiment.** To confirm the antidepressant activity of the extract of *P. eryngii*, a forced swimming test (FST) was performed, as described in the Materials and Methods. Each drug was administered by i.p. with 20 mg/kg concentration. Four mice were used in each group, and two independent experiments were performed. * *p <* 0.05, ** *p <* 0.01 indicate a significant difference compared to the control group.

## Data Availability

Not applicable.

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
