# Peer review of "Identification of the Antidepressant Function of the Edible Mushroom Pleurotus eryngii"

_jof, 2021, doi:10.3390/jof7030190_

Round 1

Reviewer 1 Report

Dear authors,Congratulations on your interesting work, which is part of the search for candidates for new antidepressants of natural origin. Does Pleurotus eryngii occur naturally in the forest environment on decaying tree trunks from where it is obtained or is it artificially cultivated? Is it commercially available as a dietary supplement.It would be good to emphasize what was new and what is a confirmation of previous observations and to speculate on the possibility of new drugs (when?). Is the consumption of the studied mushrooms currently high? and are they used in natural medicine as dietary supplements supporting traditional treatment?

The article entitled "Identification of the antidepressant function of the edible mushroom Pleurotus eryngii" raises the important issue of depression becoming more common in humans, especially in the current pandemic situation. Although there are medications but unfortunately they have their side effects, so using the edible mushroom as a dietary supplement could be an interesting alternative. I encouraged the authors to write a few words, e.g. whether the mushroom occurs naturally in forests (is it common or rare) and whether there are possibilities of growing it artificially. Or maybe it is already available in shops similarly to Pleurotus ostreatus or Lentines edodes also showing interesting cankostatic properties. I recommend the acceptance of this manuscript at current form, because I think that even without this additional information it is interesting and well written.   In my opinion, the manuscript's strengths lie in its well-designed experiments from which much experimental data was obtained. These served to draw conclusions that were fully justified. However, there is still a long way to go to exploit the properties of the fungus as a drug (maybe a weakness) requiring further research. The first step has already been taken and consumers can be encouraged to consume the mushroom (if available) as a dietary supplement.

I think it is not necessary to work further on the manuscript to improve it, the main idea in the article that we are on the right track to develop an antidepressant drug from a natural source is in itself very interesting and all the research that has been done and presented in the article.

In the discussion, we could emphasize the need to take care of biodiversity in forests because it is an invaluable resource (pharmacy) that can cure us or save our lives. It has recently been shown that even the pine pathogen Heterobasidion annosum, which was fought by foresters because it caused root and butt rot for Scots pine, has proved useful in treating colon cancer without showing any side effects. Tests were performed similarly to P. eryngii on in vitro cell lines and mice. This type of discussion (message to the public) could be given, but I do not think it is necessary.

Author Response

Thank you very much for giving good comments. I think your comments will improve my paper. Here are the answers to your comments.

Dear authors,Congratulations on your interesting work, which is part of the search for candidates for new antidepressants of natural origin. Does Pleurotus eryngii occur naturally in the forest environment on decaying tree trunks from where it is obtained or is it artificially cultivated?

  • We performed research using artificially cultivated P. eryngii

 Is it commercially available as a dietary supplement.

  • It is not commercially available now.

It would be good to emphasize what was new and what is a confirmation of previous observations and to speculate on the possibility of new drugs (when?).

  • This is very important comment and now we are testing the potential as a new drug.

 Is the consumption of the studied mushrooms currently high?

  • It is one of the most consumed mushrooms in Korea.

and are they used in natural medicine as dietary supplements supporting traditional treatment?

  • There is no report that it is used as a traditional medicine yet.

The article entitled "Identification of the antidepressant function of the edible mushroom Pleurotus eryngii" raises the important issue of depression becoming more common in humans, especially in the current pandemic situation. Although there are medications but unfortunately they have their side effects, so using the edible mushroom as a dietary supplement could be an interesting alternative. I encouraged the authors to write a few words, e.g. whether the mushroom occurs naturally in forests (is it common or rare) and whether there are possibilities of growing it artificially. Or maybe it is already available in shops similarly to Pleurotus ostreatus or Lentines edodes also showing interesting cankostatic properties. I recommend the acceptance of this manuscript at current form, because I think that even without this additional information it is interesting and well written.   In my opinion, the manuscript's strengths lie in its well-designed experiments from which much experimental data was obtained. These served to draw conclusions that were fully justified. However, there is still a long way to go to exploit the properties of the fungus as a drug (maybe a weakness) requiring further research. The first step has already been taken and consumers can be encouraged to consume the mushroom (if available) as a dietary supplement.

  • Thank you for your comments.

I think it is not necessary to work further on the manuscript to improve it, the main idea in the article that we are on the right track to develop an antidepressant drug from a natural source is in itself very interesting and all the research that has been done and presented in the article.In the discussion, we could emphasize the need to take care of biodiversity in forests because it is an invaluable resource (pharmacy) that can cure us or save our lives. It has recently been shown that even the pine pathogen Heterobasidion annosum, which was fought by foresters because it caused root and butt rot for Scots pine, has proved useful in treating colon cancer without showing any side effects. Tests were performed similarly to P. eryngii on in vitro cell lines and mice. This type of discussion (message to the public) could be given, but I do not think it is necessary.

  • Thank you for your comments. Our group will try to find the potential as a medicine from various mushrooms as you commented.

Sinerely yours,

Cheol-Won Yun

Reviewer 2 Report

The article titled Identification of the antidepressant function of the edible mushroom Pleurotus eryngii, submitted for publication by Park et al., describes the purification of tryptamine from fruiting bodies of Pleurotus eryngii.

Although the work may be interesting, the article is very poorly presented. The introduction does not provide enough information about the work's content so that the reader feels lost when entering the section of the purification of the siderophores present in the fruiting body. Authors must tell a story with a logical sequence.

The article figures are sometimes tricky to understand, and there is evidence (forced swimming, for example) whose biological significance cannot be adequately assessed by the readers of this magazine.

Unfortunately, I consider that the article needs to be deeply revised to make it appropriate and acceptable for this journal.

Author Response

Thank you very much for giving good comments. I think your comments will improve my paper. Here are the answers to your comments.

The article titled Identification of the antidepressant function of the edible mushroom Pleurotus eryngii, submitted for publication by Park et al., describes the purification of tryptamine from fruiting bodies of Pleurotus eryngii.

Although the work may be interesting, the article is very poorly presented. The introduction does not provide enough information about the work's content so that the reader feels lost when entering the section of the purification of the siderophores present in the fruiting body. Authors must tell a story with a logical sequence.

  • Thank you for your comment and we went over the manuscript as a whole. But we ask for your understanding that the current result should be completed with this manuscript. Thank you again.

The article figures are sometimes tricky to understand, and there is evidence (forced swimming, for example) whose biological significance cannot be adequately assessed by the readers of this magazine.

  • Thank you for your comment and figures were changed for easy understanding. Also we described about FST again for easy understanding (line 314-316).

Sincerely yours,

Cheol-Won Yun

Reviewer 3 Report

The article entitled ''Identification of the antidepressant function of the edible mushroom Pleurotus eryngii'' is well written and the majority of the developed methodologies are suitable described.  However, the authors must pinpoint the novelty/goal and the necessity of the current study since other species of mushrooms have already been investigated for their antidepressant functions and tryptamine (or their derivatives) exhibits well-known antidepressant activity. I recommend that the authors consider the below comments and improve their paper before publishing.

  1. Were other fractions of mushroom extract tested for their antidepressant function? If not, which were the criteria that the authors took into account for testing only the specific fraction for antidepressant activity?
  2. Is the inhibition rate of mushroom extract higher, equal or lower compared to other established SSRIs, reported in the literature?
  3. Why the authors choose to study Pleurotus eryngii and not other mushrooms species? Is this species known for the high tryptamine content?
  4. Line 82: Correct ''polar C18 RP-HPLC columns''. The stationary phase of C18 columns is non polar. 
  5. Line 87: The section ''2.3 LC-MS analysis'' does not describe a method that includes liquid chromatography, but a method that consists direct flow injection analysis using mass spectrometer as detector. Please provide info regarding the conditions of the liquid chromatography (column and instrument used, column temperature, mobile phase etc) or correct the section's title to ''FIA-MS analysis''. 
  6. According to other published works, the cell lines commonly used in depression-related studies are glial or lymphoblastoid cell lines. Why did the authors choose adenocarcinoma cell lines to a depression-oriented study? The authors should clarify if the initial goal of this work was to examine the anticancer activity of mushroom extracts/fractions.
  7. Tryptamine detection was performed only through the parent m/z ion of this compound? No MS2 analysis was not conducted? A specific fragmentation pattern and the m/z of the product ions (MS2) of a compound is normally required to reliably identify a molecule, since -especially in natural extracts- is quite possible for two different compounds to have the same precursor ion. In addition, did the authors identify other tryptamine derivatives based on their retention time and/or mass spectra?
  8. According to Figure 5, the inhibition rate of 0.01 mg/ml is higher than the inhibition rate of 1 mg/ml in both mixture and R2. However, the authors have declared that 1 mg/ml provided the higher inhibition rate. Please, provide the units in which is expressed the inhibition rate or clarify which concentration exhibited the higher antidepressant function.
  9. The paragraph concerning the tryptamine in the Conclusion fits better in the Introduction since it is the compound-of-focus of this study. 

Author Response

Thank you very much for giving good comments. I think your comments will improve my paper. Here are the answers to your comments.

The article entitled ''Identification of the antidepressant function of the edible mushroom Pleurotus eryngii'' is well written and the majority of the developed methodologies are suitable described.  However, the authors must pinpoint the novelty/goal and the necessity of the current study since other species of mushrooms have already been investigated for their antidepressant functions and tryptamine (or their derivatives) exhibits well-known antidepressant activity. I recommend that the authors consider the below comments and improve their paper before publishing.

  1. Were other fractions of mushroom extract tested for their antidepressant function? If not, which were the criteria that the authors took into account for testing only the specific fraction for antidepressant activity?

  • As described in results section, we wanted to identify the siderophores first. However, we identified molecular structure of tryptamine from the specific fraction at the final step and started the experiment with that. So, we tested fractions of the final step and went to fraction of previous stage which has activity. We did not tested all other fractions.

  1. Is the inhibition rate of mushroom extract higher, equal or lower compared to other established SSRIs, reported in the literature?

  • Our experiments showed that mushroom extract showed a little lower activity than Prozac which is an established SSRI as shown in figure 4,5, and 6.

  1. Why the authors choose to study Pleurotus eryngii and not other mushrooms species? Is this species known for the high tryptamine content?

  • As described in results section, we wanted to identify the siderophores first. However, we identified molecular structure of tryptamine from the specific fraction at the final step and started the experiment with that. And there are no reports about tryptamine of P. eryngii.

  1. Line 82: Correct ''polar C18 RP-HPLC columns''. The stationary phase of C18 columns is non polar. 

  • Thank you for your comment and we corrected it as you commented.

  1. Line 87: The section ''2.3 LC-MS analysis'' does not describe a method that includes liquid chromatography, but a method that consists direct flow injection analysis using mass spectrometer as detector. Please provide info regarding the conditions of the liquid chromatography (column and instrument used, column temperature, mobile phase etc) or correct the section's title to ''FIA-MS analysis''. 

  • Thank you for your comment and corrected the section's title to ''FIA-MS analysis'' as you commented.

  1. According to other published works, the cell lines commonly used in depression-related studies are glial or lymphoblastoid cell lines. Why did the authors choose adenocarcinoma cell lines to a depression-oriented study? The authors should clarify if the initial goal of this work was to examine the anticancer activity of mushroom extracts/fractions.

  • The reason we used adenocarcinoma cell lines is that it is reported that serotonin receptors are expressed in cancer cells a lot, and there are many reports of cell experiments using cancer cell lines to perform receptor binding assay. Therefore, for this reason, we used cancer cell lines to carry out cell silencing.

  1. Tryptamine detection was performed only through the parent m/z ion of this compound? No MS2 analysis was not conducted? A specific fragmentation pattern and the m/z of the product ions (MS2) of a compound is normally required to reliably identify a molecule, since -especially in natural extracts- is quite possible for two different compounds to have the same precursor ion. In addition, did the authors identify other tryptamine derivatives based on their retention time and/or mass spectra?

  • We identified tryptamine through the parent m/z ion and no other derivatives were tested because standard molecule of tryptamine (Sigma-Aldrich, Com) and the fraction showed same ms pattern.

  1. According to Figure 5, the inhibition rate of 0.01 mg/ml is higher than the inhibition rate of 1 mg/ml in both mixture and R2. However, the authors have declared that 1 mg/ml provided the higher inhibition rate. Please, provide the units in which is expressed the inhibition rate or clarify which concentration exhibited the higher antidepressant function.

  • Thank you for your comment. There is error in figure legend to explain inhibition rate and we changed figure.

  1. The paragraph concerning the tryptamine in the Conclusion fits better in the Introduction since it is the compound-of-focus of this study. 

  • Thank you for your comment but please understand that the story will not go well with the change, so it will be left as it was. Because we found tryptamine in the process of research and we performed experiment with that.

Sincerely yours,

Cheol-Won Yun

Round 2

Reviewer 3 Report

No further comments or suggestions are required for the revised manuscript.